

# *PTC2* region genotypes counteract *Biomphalaria glabrata* population differences between M-line and BS90 in resistance to infection by *Schistosoma mansoni*

Michael S. Blouin[1], Stephanie R. Bollmann[1] and Jacob A. Tennessen[2]

[1] Integrative Biology, Oregon State University, Corvallis, Oregon, United States of America
[2] Immunology and Infectious Diseases, Harvard T.H. Chan School of Public Health, Boston, Massachusetts, United States of America

## ABSTRACT

**Background:** *Biomphalaria glabrata* is a snail intermediate host for *Schistosoma mansoni*, a trematode responsible for human schistosomiasis. BS90 is one of the most well studied strains of *B. glabrata* owing to its high resistance to infection by most strains of *S. mansoni*. An F2 mapping study from 1999 identified two RAPD markers that associated with what appeared to be single-locus, dominant resistance by the BS90 population relative to the susceptible M-line population. One marker cannot be mapped, but the other, *OPM-04*, maps to within 5 Mb of *PTC2*, a region we recently showed has a very large effect on resistance within another snail population challenged by the same strain of parasite (PR1). Here we tested the hypothesis that the *PTC2* region contains the causal gene/s that explain the iconic resistance of BS90 snails.

**Methods:** We used marker-assisted backcrossing to drive the BS90 version of the PTC2 region (+/−~1 Mb on either side) into an M-line (susceptible strain) genetic background, and the M-line version into a BS90 genetic background. We challenged the offspring with PR1-strain schistosomes and tested for effects of allelic variation in the *PTC2* region in a common genetic background.

**Results:** Relative to M-line haplotypes, the BS90 haplotype actually confers enhanced susceptibility. So we reject our original hypothesis. One possible explanation for our result was that the causal gene linked to *OPM-04* is near, but not in the *PTC2* block that we introgressed into each line. So we used an F2 cross to independently test the effects of the *PTC2* and *OPM-04* regions in a randomized genetic background.

We confirmed that the BS90 haplotype confers increased susceptibility, and we see a similar, although non-significant effect at *OPM-04*. We discuss possible reasons why our results differed so dramatically from those of the 1999 study. We also present Pacbio assemblies of the *PTC2* and flanking region in BS90 and M-line, compare with previously published *PTC2* haplotypes, and discuss candidate genes that might be behind the enhanced susceptibility of the BS90 haplotype.

Corresponding author
Michael S. Blouin,
blouinm@science.oregonstate.edu

## INTRODUCTION

Schistosomiasis is a trematode parasitic disease of humans that affects over 200 million people in tropical regions worldwide, causing a chronic, debilitating condition that kills over 200,000 people annually (*World Health Organization, 2022*). The life cycle of schistosomes requires passage through an aquatic snail intermediate host. Humans are infected by spending time in water inhabited by infected snails. *Biomphalaria glabrata* is a New World snail that is the most well-studied intermediate host of *S. mansoni*. Resistance to *S. mansoni* is highly heritable and easily selected for in *B. glabrata* (*Richards & Shade, 1987*; *Richards, Knight & Lewis, 1992*; *Webster, Gower & Blair, 2004*; *Tennessen et al., 2015b*). So there is substantial interest in using resistance alleles to genetically alter natural snail populations to make them less able to transmit the parasite (*de Marques et al., 2014*; *Famakinde, 2018*; *Maier et al., 2019*; *Grewelle et al., 2021*), as is being developed for mosquitos (*Pham et al., 2019*; *Adolfi et al., 2020*; *Carballar-Lejarazú et al., 2020*). However, the key genes and molecular pathways underlying this genetic variation for resistance in snails are not yet well understood.

Genome-wide mapping studies using various strains of *B. glabrata* and *S. mansoni* have revealed five genomic regions in which allelic variation controls resistance to *S. mansoni* (*Knight et al., 1999*; *Goodall et al., 2006*; *Tennessen et al., 2015b*; *Tennessen et al., 2015a*; *Tennessen et al., 2020*). Here we use the terms "susceptibility" and "resistance" to mean the percentage of snails in a population that do, or do not, get infected after individuals are challenged with a fixed number of schistosome miracidia. We also use the terms "strain" and "population" interchangeably, acknowledging that most of the "strains" referred to in the literature are genetically variable.

The region showing the strongest effect on resistance to date is the *PTC2* (Polymorphic transmembrane complex 2) region. This region was identified *via* a genome-wide association study (GWAS) using outbred 1316R snails challenged by PR1 schistosomes (*Tennessen et al., 2020*). In 1316R snails the *PTC2* is a 300 to 450 kb region that contains 11 genes, nine of which code for single-pass transmembrane proteins. Haplotypes for the *PTC2* region are highly divergent in both structure and sequence. Haplotypes vary by multi-kb indels and some genes are found on only one or two of the three haplotypes. The enrichment for transmembrane proteins in this region is similar to what was found in the *GRC* (= *PTC1*) region (*Tennessen et al., 2015b*), which suggests a model in which molecular recognition (either of the parasite by the host, or vice versa) plays an important role in snail-schistosome compatibility polymorphisms (*e.g.*, *Theron et al., 2008*).

BS90 is a Brazilian strain of *B. glabrata* that is highly resistant to most strains of *S. mansoni*. Thus, it is one of the most-well-studied strains of *B. glabrata*, the subject of dozens, if not hundreds, of functional studies on its resistance over the last 50 years (*e.g.*, *Zhang et al., 2008*; *Yoshino et al., 2012*; *Ittiprasert & Knight, 2012*; *Pila et al., 2016a*, *2016b*). Therefore, there is substantial interest in finding the loci that make BS90 so much more

resistant than other strains of *B. glabrata*. *Knight et al. (1999)* performed an F2 QTL mapping cross to study the difference in resistance between BS90 and M-line strain snails to the SmPR1 and NMRI strains of *S. mansoni*. M-line is highly susceptible to SmPR1 and NMRI *S. mansoni*, while BS90 is 100% resistant to them, and the difference between snail strains appeared to segregate as a simple Mendelian trait with the BS90 allele dominant. *Knight et al. (1999)* used either SmPR1 or NMRI strain *S. mansoni* in their experiments, as the two strains behaved identically towards M-line and BS90 (M. Knight, 2022, personal communication). *Knight et al. (1999)* scored RAPD markers in the F2s and reported the sequences of two bands that associated with resistance (RAPD is an older DNA fingerprinting technique that relies on low-stringency PCR amplification using single, arbitrary, 10mer primers). After the publication of the *B. glabrata* genome it became clear that one of their bands (OPZ-11) is embedded in repetitive sequence that is dispersed throughout the genome, and so could not be localized. However, the other band (locus OPM-04) maps to a single location in the *B. glabrata* genome (*Tennessen et al., 2020*).

In our PacBio assemblies of *B. glabrata* genomes, *OPM-04* is only 5 Mb from the *PTC2* region (*Tennessen et al., 2020*). Furthermore, from our linkage map (*Tennessen, Bollmann & Blouin, 2017*) we estimated that *OPM-4* is about 20 cM from the PTC2 region, which is similar to the 17 cM recombination distance one estimates OPM-04 to be from the causal gene, based on *Knight et al. (1999)* published infection data (*Tennessen et al., 2020*). So we hypothesized that the *PTC2* region contains the actual causal gene/s linked to the *OPM-04* marker that makes BS90 so much more resistant than M-line. Here we report a test of that hypothesis, in which we used marker-assisted backcrossing to drive the BS90 version of the PTC2 region (+/−~1 Mb on either side) into an M-line genetic background, and the M-line version into a BS90 genetic background. We find that segregation of the BS90 and M-line alleles in a common genetic background has a significant effect on resistance to PR1 schistosomes, but that the BS90 allele confers increased susceptibility. So this experiment rejects the hypothesis that the PTC2 region explains the difference in susceptibility between BS90 and M-line. Furthermore, using an F2 cross we could not replicate *Knight et al. (1999)* result that a BS90 allele at *OPM-04* segregates with resistance.

## MATERIALS AND METHODS

### Snail strains

The original M-line was reportedly created in the 1950s by crossing a pigmented Puerto Rican strain that was susceptible to Puerto Rican schistosomes, with an albino, resistant Brazilian strain, and then inbreeding with selection for albinism and susceptibility (*Newton, 1955*; *Richards & Merritt, 1972*). BS90 was reportedly isolated in Salvador, Bahia, Brazil, in the 1960s, and has been in labs in the US since the 1990s (*Paraense & Correa, 1963*, cited in *Knight et al., 1999*; C. Bayne, 2006, personal communication). Both have been maintained by the Biomedical Research Institute (BRI) schistosomiasis resource center (https://www.afbr-bri.org/schistosomiasis/) and in dozens of labs around the world for decades. We obtained BS90 in 2015 from Timothy Anderson's lab at the Texas Biomedical Research Institute (originally from BRI), and M-line snails in 2018 from the BRI (their "T0" population). For our crosses we used the outbred M-line population and

an inbred line of BS90 that we generated *via* two generations of selfing (line FSS5). Snails were housed in 7.5 liter plastic tanks containing artificial spring water (*Ulmer, 1970*).

## Schistosome challenges

We challenged 6 to 9 mm diameter snails individually in 24-well plates as follows: 15 miracidia of PR1-strain *S. mansoni* were added to ~1 ml water in each well of a 24-well plate. Then a single snail was put into each well. Snails were left in the plates with miracidia for two hours, and were then moved to darkened tubs of ~24 snails per tub. Snails were kept in the darkened tubs for four weeks, and then were checked every week or two for shedding for six more weeks. Snails that did not shed within the 10 week window were classified as non-infected. Individuals that died before shedding were not included in the analysis. BS90 is always 100% resistant to PR1 (*i.e.*, no BS90 snails ever got infected), while about 60% of our M-line snails get infected under these challenge conditions. In all experimental challenges for this study we always included at least 12 pure BS90 and 12 pure M-line snails as controls for schistosome infectivity. The Oregon State University Institutional Animal Care and Use Committee provided approval for this research under Animal Care and Use Protocols 2021-0213 and 5115.

## PacBio sequencing and characterization of the *PTC2* region in BS90 and M-line

We used an inbred line of BS90 (two generations of selfing) and an outbred population of M-line for our crosses. We obtained PacBio genomic assemblies from a snail from the inbred BS90 line and from an M-line snail. The BS90 and M-line samples were run on a Sequel II machine and assembled using Flye (*Kolmogorov et al., 2019*). We generated assemblies with high contiguity for both BS90 (2,534 fragments, length = 843.8 Mb, N50 = 2.3 Mb, largest scaffold = 15.5 Mb) and M-line (1,834 fragments, length = 852.5 Mb, N50 = 15.1 Mb, largest scaffold = 38.5 Mb). In both genomes, *PTC2* occurs on assembled fragments over 4 Mb long, more than 1 Mb from either end, allowing for detailed assessment of the surrounding genomic region. We used alignments of the BS90 and M-line sequences to find regions of synteny around the *PTC2* region in which we could place PCR markers for the marker-assisted backcrossing (Fig. 1). We used GENSCAN (*Burge & Karlin, 1997*), HMMER (*Finn, Clements & Eddy, 2011*), DeepTMHMM (*Hallgren et al., 2022*), Vectorbase (*Giraldo-Calderón et al., 2015*), and BLASTP to annotate coding regions in these assemblies, supplemented by our previous genomic and transcriptomic data from the 1316R population of *B. glabrata* (*Tennessen et al., 2020*). Assemblies have been made available through NCBI Genome, BioProject Accession PRJNA847915.

## Backcross experiment

We used PCR to score five markers in and around the *PTC2* region (Fig. 1). For ease of scoring, we designed primers to target gel-scorable indels that differed between our M-line and BS90 parental strains (Table S1). To drive the BS90 *PTC2* region into an M-line genetic background we created F1s, and then backcrossed them to pure M-line ("B into M"
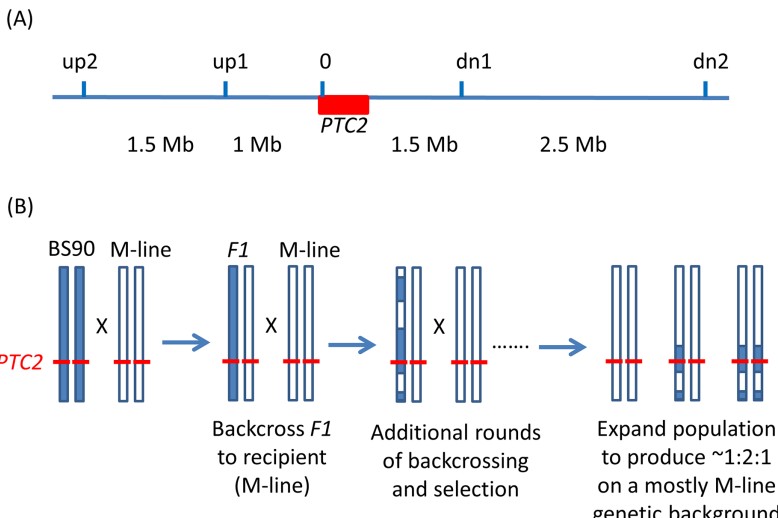

**Figure 1 Backcrossing scheme.** (A) Relative positions of the markers used in marker-assisted back-crossing (actual primer locations are in Table S1). Red box = the *PTC2* region, as defined in *Tennessen et al. (2020)*. Marker *0* is in, but near the far upstream end of the *PTC2* region. Distances indicated between markers are approximate. *Knight et al. (1999)* RAPD marker *OPM-04* is approximately 2.5 Mb upstream of *up2*. (B) Illustration of driving the BS90 haplotype into an M-line genetic background. We looked for recombinants between one set of exterior markers (*e.g.*, between *up2* and *up1*). We then backcrossed those individuals and their offspring repeatedly to pure M-line snails until we obtained a crossover between the other pair of exterior markers (*e.g.*, between *dn1* and *dn2*). Then we expanded tanks that were started using the offspring of an MHHHM individual (heterozygous for the three interior markers, but pure M-line at the flanking markers), and then tested for effects of segregation at *PTC2* in the mostly M-line genetic background. We created four of these "B into M" tanks. The same approach was used to produce a single tank segregating for the *PTC2* haplotypes in a mostly BS90 genetic background. We refer to the block of genome between *up2* and *dn2* as the "backcross region", and use "*PTC2* region" to refer just to the ~300 Kb region described in *Tennessen et al. (2020)* = Red box.

backcrossing). We genotyped backcross offspring until we found recombinants inside one of the outer two markers, and then used those offspring in repeated rounds of backcrossing and genotyping until we found individuals that were homozygous M-line at the two outer markers and heterozygous at the inner three markers (*i.e.*, genotype MHHHM at the five loci, where M = homozygous for M-line allele and H = heterozygous; Fig. 1). This took three or four rounds of backcrossing per tank. Individual snails with the desired double recombination event (MHHHM) were backcrossed one final time to further remove background BS90 genome and to produce a large number of siblings which all had the MHHHM genotype. We then let tanks of MHHHM siblings breed randomly and expand to create populations containing MMMMM, MHHHM and MBBBM individuals on a randomized genetic background for challenge testing. Each tank was started from an independent double recombinant event. We created and tested four independent "B into M" populations (*i.e.*, started by MHHHM individuals), named M8.2, M8-2.2, M13.2 and M17. We challenged and genotyped N = 159, 89, 85 and 68 surviving snails from each of these populations, respectively.

We used the same genotyping and backcross approach to drive the M-line *PTC2* region into a BS90 genetic background ("M into B" backcross population). We were able to find

only a single BHHHB individual (where B = homozygous for the BS90 allele). We used that snail to create a single "M into B" population (population B7.2), following the same protocol as with the "B into M" populations. B7.2 produced only 11 infected individuals out of 116 challenged survivors (9.2%).

## F2 Intercross population to test regional effect

We also crossed BS90 × M-line to create an F2 cross population that we allowed to breed randomly for almost two years while we conducted the backcrossing experiments (this population was created as a backup mapping population in case we rejected the *PTC2* hypothesis). We pooled F1s from two individual BS90 × M-line individual snail crosses, and then used their offspring to found the F2 breeding population. Our lab was abruptly shut down during the covid pandemic, during which time we were able to maintain the F2 breeding population. However, we lost the tubs that contained the original founding parents, so we don't have their DNA. The BS90 parents were all from an inbred line, so are likely to be similar to the source inbred line, which we retain and for which we have a PacBio assembly. As with the backcross populations, M-line parents were from our outbred population. So there could be genetic variation within the M-line contribution to the F2 population.

Seven months after setting up the F2 population, we started challenging ~two dozen snails every few months until we had accumulated >300 phenotyped individuals. We challenged a total of 424 individuals, of which 333 survived and 62 (19%) became infected. We successfully genotyped 329 of these 333 individuals at the five loci that span the backcross region. We chose 55 infected and 55 non-infected individuals as cases and controls, and additionally genotyped them *via* PCR at *OPM-04* (primers in Table S1) in order to compare the effects of allelic variation at *OPM-04 vs PTC2*. The *OPM-04* amplicons were scored *via* Sanger sequencing. In the full sample of 333 individuals we observed evidence of null alleles at marker loci *0*, *dn1* and *dn2*, while markers *up1* and *up2* showed Hardy-Weinberg Equilibrium. So we analyzed genotypes at marker *up1* as a proxy for *PTC2* genotype (only ~1 Mb away, so should be in high linkage disequilibrium with marker *0*). We noticed no evidence for null alleles in the backcross populations, but that is expected because each generation we used individuals that were successfully typed at all five loci.

## Statistical analysis

The various populations created, and samples used, for this study are summarized for clarity in Fig. S1. The effect of segregation for BS90 *vs* M-line alleles on infection status in the backcross and F2 populations was analyzed *via* binary logistic regression in Systat 13.2. For the four "B into M" backcross populations, we tested main effects of genotype and population, and the genotype × population interaction. We tested additive and completely dominant modes of inheritance. We observed too few infected snails from the "M into B" population to have any power for a statistical test, so we simply show the data from B7.2 and note that the general pattern is consistent with what we observed in the "B into M" populations.

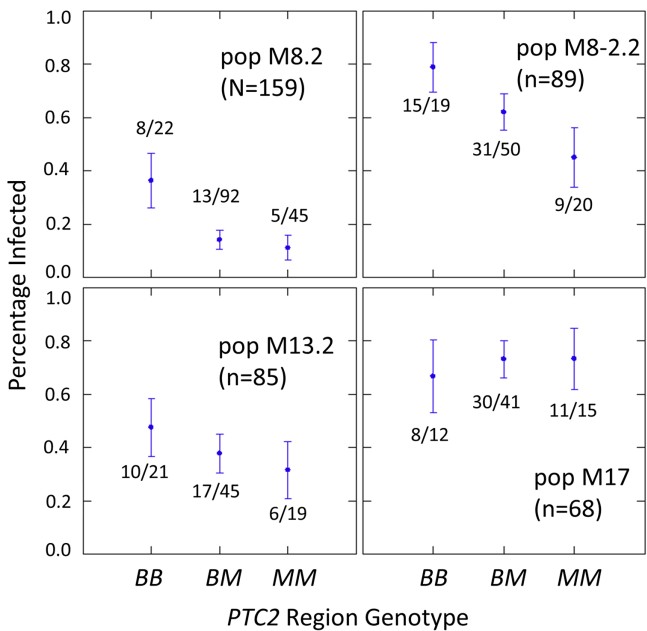

**Figure 2 Effect of *PTC2* region (locus *0*) genotypes in four different backcross-into-M genetic backgrounds.** *B* = BS90 allele. *M* = M-line allele. Number of snails challenged and genotyped for each population are in parentheses. Number of snails infected per number of that genotype are also labelled. Error bars = standard error of the proportion. The population and genotype effects were significant, but the population-by-genotype interaction effects were not. Estimates and *P*-values for each parameter in the full model are in Table S3. For the genotypic effect, the additive and M-allele-dominant models had similar support and gave odds ratios of 2.2, *P* = 0.025 and 3.8, *P* = 0.009, respectively.

We tested the effect of *PTC2* genotypes using marker *0* in the backcross populations, and using marker *up1* in the F2 population (to avoid the null allele at marker *0*). For locus *OPM-04*, we observed two sequences from the M-line population segregating in the F2s, both of which were distinct from the BS90 sequence (Fig. S2). So we analyzed those data as bi-allelic, *B vs M* (where *M* = either of the two M-line alleles).

## RESULTS

### Effect of *PTC2* in mostly M-line genetic backgrounds

The number of infected and not-infected individuals per *PTC2* genotype in each "B into M" backcross population is shown in Fig. 2. The additive model showed only a slightly better fit than an M-allele-dominant model (AIC = 458.8 *vs* 459.6). There were strong main effects of both genotype and population on infection status (Tables S2, S3). The genotype-by-population interactions were not significant, despite the visual appearance of a different genotypic effect in population 17 in Fig. 2. Under an additive model, each additional BS90 allele roughly doubles the odds of infection (odds ratio ± standard error = 2.2 ± 0.78; *P* = 0.025). The estimated effect is similar under an M-allele dominant model (odds ratio = 3.8 ± 1.9 for the *BB* genotype relative to *M_* = *MM* or *BM*; *P* = 0.009).
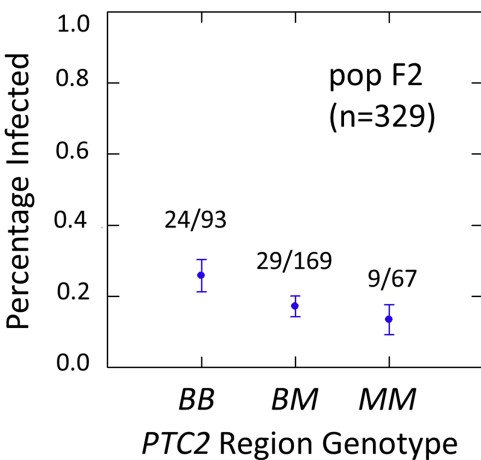

**Figure 3 Effects of *PTC2* region (locus *up1*) in the full sample from the F2 population (N = 329).**
*B* = BS90 allele. *M* = M-line allele. Number of snails infected per number of that genotype are label-led. Error bars = standard error of the proportion. The genotype effect was significant, with odds ratios = 1.5, *P* = 0.041 and 1.8, *P* = 0.054 for additive and dominant models, respectively. The additive and M-allele-dominant models were equally supported.

One unexpected result is that we see a significant heterozygote excess at the *PTC2* in population M8.2, and a trend of heterozygote excess in the other three "B into M" populations ($F_{IS}$ at locus $0 = -0.18$, $-0.12$, $-0.06$ and $-0.21$ in populations M8.2, M8-2.2, M13.2 and M17, respectively). Combining all four populations together, observed heterozygotes ($N = 228$) significantly exceed expectation ($\chi^2$ test, $P = 0.003$), whether expected heterozygosity is calculated from each population separately (expected $N = 198.7$) or from all samples pooled (expected $N = 199.7$). We see similar negative $F_{IS}$ at all three of the segregating backcross-region loci (*up1*, *0* and *dn 1*) in these populations, so the negative $F_{IS}$ seems unlikely to result from technical artifacts.

### Effect of *PTC2* in a mostly BS90 genetic background

We see infection in all three genotypes, and the trend is consistent with what we observed in the four "B into M" populations (fractions infected were 3/33 = 0.11 *BB*, 7/64 = 0.11 *BM*, and 1/19 = 0.06 *MM*). Population B7.2 also showed a trend of heterozygote excess at the backcross region loci ($F_{IS} = -0.12$ at all three loci), although the deviation from Hardy Weinberg Equilibrium is not significant.

### Effect of *PTC2* in the F2-cross genetic background, and comparison with *OPM-04*

We examined effects of the *PTC2* and *OPM-04* regions in the F2 population in order to test whether the locus linked to *OPM-04* in *Knight et al. (1999)* was still nearby, even if not within the region we introgressed into our backcross lines. As in the backcross populations, the BS90 allele conferred significantly enhanced susceptibility in the full sample from the F2 population ($N = 329$ successfully genotyped) (Tables S2, S3, Fig. 3). The additive and M-allele-dominant models were equally supported. Odds ratios for increased susceptibility

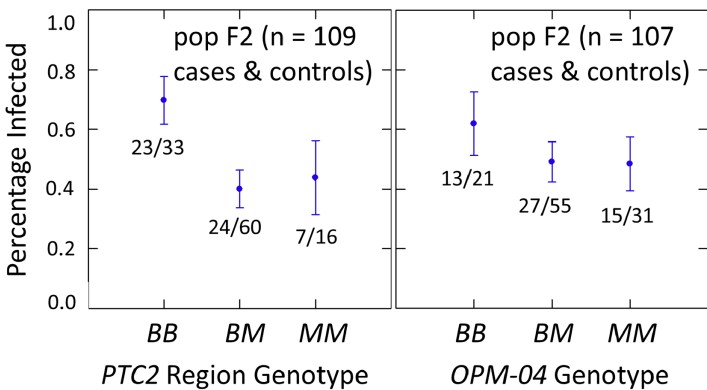

**Figure 4** **Effects of *PTC2* region (locus *up1*) and locus *OPM-04* in the sub-sample of 55 cases and 55 controls.** Total sample size is slightly less than 110 for each locus owing to PCR failures. *B* = BS90 allele and *M* = M-line allele. Number of snails infected per number of that genotype are labelled. Error bars = standard error of the proportion. The genotypic effect was significant for *PTC2* but not for *OPM-04*. For *PTC2*, the M-allele-dominant model had a better fit (AIC = 147.25 *vs* 150.05). The genotypic effect gave odds ratios of 2.0, *P* = 0.029 and 3.3, *P* = 0.007 for additive and dominant models, respectively. The average infection rate is higher than in Fig. 3 because the 329 individuals in Fig. 3 were a random sample from the population.

were 1.5 (± 0.32) per *B* allele under an additive model (*P* = 0.041) and 1.8 (± 0.54) for *BB* relative to *M_* under an M-allele dominant model (*P* = 0.044).

The subsample of 55 cases and 55 controls also showed significantly enhanced susceptibility caused by the *PTC2 B* allele. Odds ratios = 2.0 (± 0.61; *P* = 0.029) and 3.3 (± 1.49; *P* = 0.007) for the genotype effect under additive and M-allele dominant models, respectively (Tables S2, S3). There was a similar pattern at *OPM-04* (Fig. 4), although the genotype effect was not significant (*P* = 0.286 and 0.380 for additive and M-allele dominant models, respectively; Tables S2, S3). So unlike our original expectations from *Knight et al. (1999)*, *OPM-04* does not appear to be anywhere near a locus at which the BS90 allele confers dominant resistance. Instead, it shows a pattern similar to that of the closely-linked *PTC2* region.

As mentioned previously, there were obvious null alleles at loci *0* to *dn2*. But loci *up2* and *up1* were in Hardy-Weinberg Equilibrium ($F_{IS}$ = −0.03), so we do not see the pattern of heterozygote excess that was apparent in the backcross populations.

## PacBio assemblies

In the 1316R population there are three *PTC2*-region haplotypes that are ordered from most to least protective as *R* > *S2* > *S1* (*Tennessen et al., 2020*). Our M-line haplotype is over 99% similar to the previously described *S2* haplotype, with a total of only six nonsynonymous differences between M-line and *S2* across all 11 genes in the region. Our BS90 haplotype is distinct from other known haplotypes; a portion of it is similar to *S2*/M-line, while the remainder is closer to *S1* and is extremely divergent from *S2*/M-line (Figs. 5, 6). At the amino acid level, *PTC2* genes 1–5 are 95% identical between BS90 and *S2*/M-line, while the remaining genes are either only present in BS90 (gene 7), only present in M-line (genes 6 and 8), or in the case of gene 9 (*BGLB018951*) present in both but over

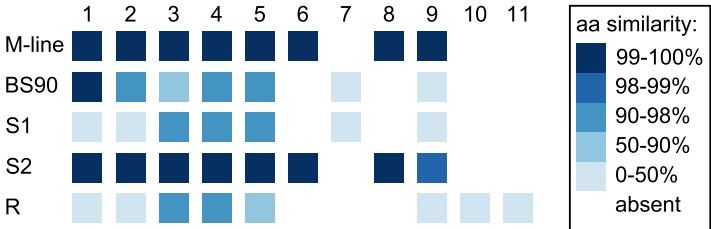

**Figure 5 Divergence among *PTC2* coding genes.** Genes are numbered 1–11 as in *Tennessen et al. (2020)*. For M-line, BS90, and the three 1316R haplotypes (*S1*, *S2*, and *R*), squares indicate which genes are present. The *R* haplotype was previously reported to not encode Gene 3 due to lack of observed expression (*Tennessen et al., 2020*), though an orthologous and syntenic open reading frame does exist, which we indicate here. Color indicates amino-acid similarity to M-line. M-line is nearly identical to *S2*, while BS90 is a mix of *S1*-like and *S2*-like sequence. The region containing genes 6–9 seems the most likely to explain the M-line *vs* BS90 difference.

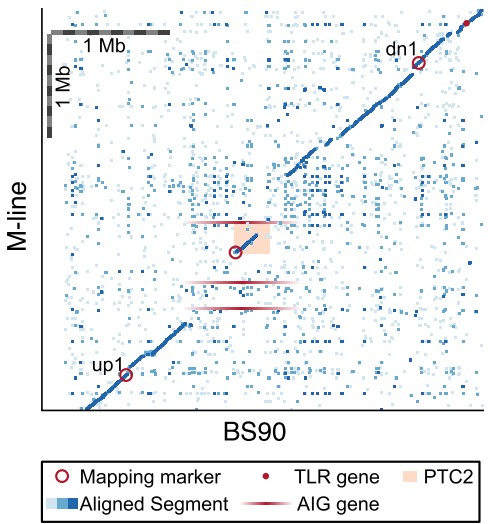

**Figure 6 Dot plot for our BS90 and M-line haplotypes in the vicinity of the *PTC2* region.** Peach rectangle = *PTC2* region identified in *Tennessen et al. (2020)*. Each dot represents a 600 bp segment, and color indicates sequence similarity (light blue = 50–90% similarity; medium blue = 90–97.5% similarity; dark blue = 97.5–100% similarity). Mapping markers are indicated (open red circles); *up2* and *dn2* lie beyond the borders of the plot. Candidate genes outside of *PTC2* include a toll-like receptor (TLR) gene and M-line AIG genes (shown with ambiguous x-axis position since they don't occur in BS90). Scale bars of 1 Mb are indicated. In the central ~1 Mb region of this plot, BS90 and M-line share almost no sequence similarity except for the first half of the *PTC2*. Gene(s) in the unshared regions are interesting candidates to explain the phenotypic differences observed between haplotypes.

50% divergent in sequence between them. Thus, this divergent portion of *PTC2* seems most likely to explain the observed phenotypic difference between BS90 and M-line.

Recall, however, that the ~300 kb *PTC2* region is embedded within a larger region (~850 kb in BS90 and ~1,300 kb in M-line) that shows almost no sequence similarity between our M-line and BS90 assemblies (Fig. 6). Thus, it is possible that the causal gene/s lie in the flanking regions, which we also introgressed. Both haplotypes possess unique genes in this zone, although as with many snail genes, little is known about their function and most

appear to be transposable elements. Notably, three AIG genes occur in this region in M-line but not BS90 (in possession of an AIG1 domain; homologs of each other and of *BGLB016854*, *BGLB012732*, and *BGLB004870*). AIG genes are suspected to have immune function in *Biomphalaria* (*Lu et al., 2020*). We have confirmed with RNA-Seq data from haplotype *S2* (Accession SRX8534561) that all three AIG1 genes are expressed. Farther out from *PTC2*, our BS90 and M-line assemblies are again similar in sequence and so seem less likely to contain the causal genes, though this remains a possibility. For example, a toll-like receptor gene *BGLB025483* occurs near locus *dn1* in both populations and has 24 amino acid differences between the haplotypes. *Pila et al. (2016b)* showed that knocking down expression of a different Toll-like receptor makes BS90 more susceptible, so this is an interesting class of proteins.

One interesting observation from our BS90 and M-line PacBio assemblies is that the 1.2 kb *OPM-04* amplicon sequence published by *Knight et al. (1999)* (genbank AF078109) is almost the same in our BS90 assembly, but is interrupted by a ~1.7 kb insertion in our M-line assembly (Fig. S2). So if *Knight et al. (1999)* M-line and BS90 also differed at this indel, then that might explain why under a low-stringency RAPD-PCR reaction they got amplification from BS90 but not from M-line chromosomes, resulting in a dominant marker for BS90.

## DISCUSSION

We have conclusively rejected the hypothesis that the *PTC2* region explains the difference between our M-line and BS90 populations in resistance to PR1-strain *S. mansoni*. Nor could we replicate *Knight et al. (1999)* result that the *OPM-04* locus was linked to a dominant resistance from BS90. Furthermore, in a common genetic background, the BS90 allele actually conferred increased susceptibility relative to the M-line alleles. This is the opposite of what was predicted, given our BS90 remains 0% susceptible to PR1 schistosomes, while our M-line is 60% susceptible.

### Why were we unable to replicate Knight et al. (1999) result?

We were surprised to see no effect of the OPM-04 region in our BS90 × M-line F2 population. Possible explanations for this failure to replicate *Knight et al. (1999)* result include: (1) the strains of "BS90", "M-line" or schistosome we used differ from theirs, and (2) we were not measuring the same phenotype.

(1) BS90 and M-line have been in culture in many labs for decades, so there is no guarantee that our lines are genetically the same as the those used by *Knight et al. (1999)* over 20 years ago. BS90 is phenotypically consistent in that it remains 0% susceptible to PR1 schistosomes. On the other hand, *Knight et al. (1999)* reported that their M-line was 100% susceptible to PR1, whereas under our experimental conditions M-line was only ~60% susceptible. Thus, our outbred M-line population may contain variation for resistance. Perhaps their M-line was more isogenic and carried a susceptibility allele near *OPM-04* that is absent or in low frequency in our M-line. Another possibility is that effects of genetic background change the penetrance of whatever allele was linked to *OPM-04* in

their M-line. It is also possible that the SmPR1 strain of schistosome we used differs in some way from the strains they used.

(2) Another plausible explanation for the difference in results between the two studies is that we were not measuring the same phenotype. In *Knight et al. (1999)* study they report phenotype ratios in BS90 × M-line F1s and F2s that are consistent with single-locus Mendelian inheritance, and in which the BS90 allele is dominant and confers complete resistance (*i.e.*, 0% infected F1s, ~25% infected F2s). Older literature on crosses among strains that differed in resistance to PR1 reported data consistent with single-locus inheritance when using mature snails, but data more consistent with polygenic inheritance when using juvenile snails (*e.g.*, *Richards, 1984*). Thus, it is possible that our experimental conditions (*e.g.*, snail size or maturity, number of miracidia per challenge, or some other environmental factor) reduced the importance of whatever highly penetrant genotype was linked to *OPM-04* in *Knight et al. (1999)* study. For example, although both studies used similar-sized snails, *Knight et al. (1999)* challenged each snail with 100 miracidia, whereas we challenged with only 15 miracidia (because higher numbers caused excess mortality in our hands).

There is no obvious reason to doubt the data in *Knight et al. (1999)*. Plus, it seems a remarkable coincidence that in a 0.9 Gb genome of 18 chromosomes, such an important region as *PTC2* would lie only 5 Mb, and almost exactly the expected number of cM from *OPM-04*, yet not contain the causal gene/s (*Tennessen et al., 2020*). Therefore, we suggest that if one could reproduce the highly-penetrant, dominant BS90 phenotype observed by *Knight et al. (1999)* and previous workers, then it would be worth re-testing whether the OPM-04 region shows an association, and whether the causal gene/s linked to that marker lie in the *PTC2* region.

## Genetic background effects

The sizable population effects among the four "B into M" populations were stronger than the genotype effect (Fig. 2; Table S3), and presumably reflect their different genetic backgrounds. This shows that other loci have a very large effect on resistance in this system. Each population was started using different M-line individuals from the outbred M-line, and each must retain different residual segments of donor line (BS90) genome that is still segregating. So there is plenty of opportunity for these populations to differ in genetic background, and we don't know to what extent that results from variation in the base M-line population or from different retained segments of BS90 chromosomes. The population-by-genotype interactions were not significant, although the trend towards a weaker genotypic effect in population M17 was intriguing (Fig. 2). The effect of SOD1-region alleles on resistance appears to depend on genetic background (*Bonner et al., 2012*), so similar epistatic effects would not be surprising.

## Negative $F_{IS}$ in the *PTC2* region

It was intriguing that $F_{IS}$ in the *PTC2* region was negative in all of the backcross populations, and significantly so in one of them, and in the set of all populations combined. The mortality rate after parasite challenge in these populations ranged from 6% to 31%,

and if these deaths were primarily *BB* and/or *MM* homozygotes, that could entirely explain the observed heterozygote excess. However, observed mortality was not correlated with $F_{IS}$ (for example, population 13.2 had the highest mortality but the lowest heterozygote excess) and heterozygotes did not show any overdominant effect on odds of infection (Figs. 2, 3). So if there is overdominance for survival, it is not obviously owing to the schistosome challenges. Unfortunately we did not keep the backcross populations at the end of the experiment, so we cannot go back and test for Hardy-Weinberg equilibrium in the unchallenged populations. A slightly negative $F_{IS}$ can also result when a small effective number of breeders ($N_b$) contributes to a sample, but to get $F_{IS}$ in the observed range of −0.15 to 0.20 would require $N_b$ ~2–3 individuals (*Waples, 2015*). Such a small $N_b$ is not out of the question, but seems low. That there might be overdominance for survival in a region of such obvious immune relevance is a hypothesis worthy of further exploration, as this would be one mechanism that could maintain genetic diversity in the region.

## Additional insights into the *PTC2* region

Allelic variation in the *PTC2* region has the largest known effect on resistance identified to date (*Tennessen et al., 2020*). The extreme structural and non-synonymous sequence variation in this region, and its enrichment for membrane-spanning protein genes, strongly suggests host-parasite interacting molecules under selection. Of course, schistosomes are an introduced species in the New World, so this region no doubt evolved with other pathogens. It will be important to determine which loci are involved in interactions with schistosomes, and if multiple loci are involved, how they interact. The 2- to 4-fold effect of *BB* vs *MM* observed here was weaker than previously reported for *PTC2* (*Tennessen et al., 2020*), but comparable to the effect of other important *B. glabrata* resistance loci like *sod1* or *RADres* (*Tennessen et al., 2015a*). Such effects are real and likely to be biologically meaningful, but they are easily missed. Here we benefitted from the use of hundreds of snails. A power analysis shows that if we had used half the sample size in this study, we likely would not have inferred a significant genotype effect. Thus lower-powered studies might miss these more subtle genotype-phenotype correlations.

Although our addition of two new assemblies of the region revealed new structural variants and additional sequence variation, the two new haplotypes closely resemble the other known haplotypes or recombinants of them. Comparing the two new haplotypes to the three previously described haplotypes allowed us to make some *a priori* predictions about which loci are most likely to be involved. For example, the M-line haplotype closely resembles *S2* while the BS90 haplotype is closer to *S1*, and thus the 7-fold decrease in infection odds conferred by *S2S2* relative to *S1S1* (*Tennessen et al., 2020*) could have the same genetic basis as the effect reported here, perhaps due to one of genes 6 through 9 or an AIG gene. We are currently attempting functional knockdowns of these genes to provide definitive tests.

## CONCLUSIONS

The most striking result of this work is that not only did the block containing *OPM-04* and *PTC2* not contain the dominant-resistance allele predicted by *Knight et al. (1999)*, the BS90

version actually conferred susceptibility. This antagonistic effect must be completely counteracted by other loci that remain to be identified, as our BS90 remains 100% resistant to PR1 schistosomes, and M-line highly susceptible. It will be extremely interesting to find those additional loci and to finally understand how they interact with whatever locus *Knight et al. (1999)* mapped over 20 years ago. Deciphering this complex, polygenic basis for resistance will be facilitated by the emerging plethora of *Biomphalaria* genome assemblies, including those presented here and others (*e.g.*, *Nong et al., 2022*). In turn, these discoveries will enable new options for vector-based disease control.

## ACKNOWLEDGEMENTS

PacBio library preparation and/or PacBio sequencing was done by the Center for Quantitative Life Sciences Core Lab at Oregon State University, and the University of Oregon Genomics and Cell Characterization Core Lab. M-line snails and SmPR1 schistosomes were provided by the NIAID Schistosomiasis Resource Center of the Biomedical Research Institute (Rockville, MD) through NIH-NIAID Contract HHSN272201700014I. Ed Davis helped with use of Flye. We thank Chloe Hanson, Maelyn Estrada, Marisa Bennett, Isabelle Ray, Abigail LaVerdure and Havana Soler for help with snail husbandry.

### Funding

This work was supported by NIH NIAID grant R01AI143991. There was no additional external funding received for this study. The funders had no role in study design, data collection and analysis, decision to publish, or preparation of the manuscript.

### Grant Disclosures

The following grant information was disclosed by the authors:
NIH NIAID: R01AI143991.

### Competing Interests

The authors declare that they have no competing interests.

### Author Contributions

- Michael S. Blouin conceived and designed the experiments, analyzed the data, prepared figures and/or tables, authored or reviewed drafts of the article, and approved the final draft.
- Stephanie R. Bollmann conceived and designed the experiments, performed the experiments, analyzed the data, prepared figures and/or tables, authored or reviewed drafts of the article, and approved the final draft.
- Jacob A. Tennessen conceived and designed the experiments, analyzed the data, prepared figures and/or tables, authored or reviewed drafts of the article, and approved the final draft.

## Animal Ethics

The following information was supplied relating to ethical approvals (*i.e.*, approving body and any reference numbers):

The Oregon State University Institutional Animal Care and Use Committee provided approval for this research under Animal Care and Use Protocols 2021-0213 and 5115.

## DNA Deposition

The following information was supplied regarding the deposition of DNA sequences:

Pacbio assemblies from the M-line and BS90 populations are available at NCBI Bioproject: PRJNA847915; SAMN28962998 and SAMN28962999.

## Data Availability

The code and raw data are available in the Supplemental Files.

## Supplemental Information

Supplemental information for this article can be found online at http://dx.doi.org/10.7717/peerj.13971#supplemental-information.

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
