# Peer review of "PTC2 region genotypes counteract Biomphalaria glabrata population differences between M-line and BS90 in resistance to infection by Schistosoma mansoni"

_PeerJ, doi:10.7717/peerj.13971_

## Round 0.1 · original submission · Minor Revisions

The review process is now complete, and two reviews are included at the bottom of this letter. All reviewers and I agree that your manuscript deserves to be published. While the experimental design and findings are original and relevant to the field, we identified some points that must be considered in your resubmission. Please, adjust the figures accordingly to the suggestions raised by the reviewers.

Reviewer 1 ·

Basic reporting

In this study, Dr. Blouin and colleagues have used marker-assisted back-crossing to drive the PTC2 region of BgBS90 snails (resistant to PR1 Schistosoma mansoni population) into an M-line snail genetic background (susceptible population to PR1 S. mansoni). They have challenged the offspring of these back-crosses and tested for effects of allelic variation in the PTC2 region in a common genetic background. Very interestingly (and intriguingly!), BS90 haplotype, instead of enhancing snail resistance to schistosome parasite, confers enhanced susceptibility, which is not what was found by Knight et al., 1999 where they shown that this BS90 allele segregated with resistance.

Experimental design

Overall, I really enjoyed reviewing this manuscript: it is well written and clear, the design of the study presented is neat.
I have only few minor comments, mainly about the Materials & Methods section or related to the quality of Figures, which could be improved (please see below).
* Materials & Methods:
- l. 163: Could the authors provide the BioProject accession number?
- l. 184-187: Did the authors performed 4 replicate tanks as well for their "M into B" population of snails?

- Data availability
For the sake of rigor and reproducibility, I would encourage authors to also share their code (used to analyze their data and generate the Figures) and submit it to repository like GitHub, Zenodo etc.

* Figures and Figure legends:
Overall, I think the quality of the Fig 1 has to be improved.
- Fig 1A: Could the authors put the loci position under the loci name ticks?
- Fig 1B: Could the authors align all the chromosome schemes horizontally and space the chromosome pairs evenly? Same remark for the "X" between chromosome pairs, they should be aligned horizontally and spaced evenly.
- Fig 2, Fig 3 and Fig 4: Could the authors display the statistical analysis on each graph (stars or letters) to facilitate the identification of statistically different vs. similar groups? Perhaps also add a summary of the statistical analysis in the Figure legends.

Validity of the findings

Very robust discussion of the results obtained and potential explanations regarding the fact that they were not able to replicate Knight et al.'s results.

Additional comments

Very interesting and clear work! It seems that snail resistance to schistosome is much more complex than expected.

Reviewer 2 ·

Basic reporting

No comment

Experimental design

No comment

Validity of the findings

No comment

Additional comments

This is a co-review by a graduate student and their graduate mentor. This manuscript clearly outlines the authors’ attempt to backcross the BS90 strain of Biomphalaria glabrata into a more susceptible strain and an overview of plausible explanations for their unexpected results. In our opinion, the paper is well written, the experimental designs are solid, and the data was carefully collected and analyzed. The results and interpretation presented here will be useful both to the Biomphalaria community and to the larger group of scientists interested in the genetic basis of infection susceptibility/resistance and connecting phenotypes to genotypes. The study also provides a nice example of the critical importance of attempting to replicate key results, and that failure to replicate does not mean that the science is not solid, but instead, that the story is more complicated.

The authors were challenged by unexpected results, but they thoroughly investigated possible reasons their initial hypothesis was rejected and that earlier experimental outcomes were not replicated. They also provided a clear path forward in the steps they are now taking to understand the discrepancies between their study and those that came before. We did not identify any major problems with the manuscript. We provide some minor suggestions for the authors below.


Specific Comments:

Line 163: Make sure you correct the BioProject Accession ID from “XXXXXXX” to the actual number.

In figure 3, the numbers above the error bars are the number of snails per genotype, but the y-axis describes percentage infected. This could be confusing.

Line 138: Consider explaining specifically what “challenging” the snails means.

At times, the language used feels too informal (line 197, 292).

---

## Round 0.2 · accepted · Accept

The authors addressed all of the comments and points raised by the reviewers accordingly.

Reviewer 1 ·

Basic reporting

After receiving and reviewing the revised manuscript, I think it is now suitable for publication. The authors carefully answered to the few comments/critics I have made and amended their manuscript accordingly. The manuscript is very clear, with neat Figures!

Experimental design

No specific comments for this second round of review on this section.

Validity of the findings

No specific comments for this second round of review on this section.

Additional comments

No specific comments for this second round of review on this section.

Reviewer 2 ·

Basic reporting

No comment

Experimental design

No comment

Validity of the findings

No comment.

Additional comments

I very much appreciate seeing careful exploration of unexpected "negative" results. We need more such examples like this in the peer-reviewed literature.